# Circumsporozoite Protein of *Plasmodium berghei*- and George Baker Virus A-Derived Peptides Trigger Efficient Cell Internalization of Bioconjugates and Functionalized Poly(ethylene glycol)-*b*-poly(benzyl malate)-Based Nanoparticles in Human Hepatoma Cells

**DOI:** 10.3390/pharmaceutics14040804

**Published:** 2022-04-06

**Authors:** Elise Vène, Kathleen Jarnouen, Catherine Ribault, Manuel Vlach, Yann Verres, Mickaël Bourgeois, Nicolas Lepareur, Sandrine Cammas-Marion, Pascal Loyer

**Affiliations:** 1Institut NUMECAN (Nutrition Metabolisms and Cancer), Inserm, UMR-S 1241, INRAE UMR-A 1341, Univ Rennes, F-35000 Rennes, France; elise.vene@chu-rennes.fr (E.V.); kathleen.jarnouen@live.fr (K.J.); catherine.ribault@univ-rennes1.fr (C.R.); manuel.vlach@agrocampus-ouest.fr (M.V.); yann.verres@univ-rennes1.fr (Y.V.); 2Pôle Pharmacie, Service Hospitalo-Universitaire de Pharmacie, CHU Rennes, F-35033 Rennes, France; 3INRAE, Institut AGRO, PEGASE UMR 1348, F-35590 Saint-Gilles, France; 4CRCINA, Inserm, CNRS, Université de Nantes, F-44000 Nantes, France; mickael.bourgeois@univ-nantes.fr; 5ARRONAX Cyclotron, F-44817 Saint Herblain, France; 6Comprehensive Cancer Center Eugène Marquis, F-35000 Rennes, France; 7Institut des Sciences Chimiques de Rennes (ISCR), Ecole Nationale Supérieure de Chimie de Rennes, CNRS UMR 6226, University of Rennes, F-35042 Rennes, France

**Keywords:** hepatoma cells, macrophages, George Baker (GB) virus A10-9, circumsporozoite protein of *Plasmodium berghei*, polymeric nanoparticles, bioconjugates, opsonization

## Abstract

In order to identify the peptides, selected from the literature, that exhibit the strongest tropism towards human hepatoma cells, cell uptake assays were performed using biotinylated synthetic peptides bound to fluorescent streptavidin or engrafted onto nanoparticles (NPs), prepared from biotin-poly(ethylene glycol)-*block*-poly(benzyl malate) (Biot-PEG-*b*-PMLABe) via streptavidin bridging. Two peptides, derived from the circumsporozoite protein of *Plasmodium berghei*- (CPB) and George Baker (GB) Virus A (GBVA10-9), strongly enhanced the endocytosis of both streptavidin conjugates and NPs in hepatoma cells, compared to primary human hepatocytes and non-hepatic cells. Unexpectedly, the uptake of CPB- and GBVA10-9 functionalized PEG-*b*-PMLABe-based NPs by hepatoma cells involved, at least in part, the peptide binding to apolipoproteins, which would promote NP’s interactions with cell membrane receptors of HDL particles. In addition, CPB and GBVA10-9 peptide–streptavidin conjugates favored the uptake by hepatoma cells over that of the human macrophages, known to strongly internalize nanoparticles by phagocytosis. These two peptides are promising candidate ligands for targeting hepatocellular carcinomas.

## 1. Introduction

In oncology, the development of most chemotherapies relies on the identification of small molecules, targeting altered molecular pathways in cancer cells, but rarely considers a site-specific delivery. Consequently, many cancer chemotherapeutics do not accumulate specifically at sites of the tumor, but distribute evenly throughout the body, often resulting in deleterious side effects and limited bioavailability. These drawbacks significantly contribute to the retrieval of promising molecules and lack of efficacy of approved drugs [1,2]. To overcome these limitations, the concept of “vectorized therapy” has emerged to achieve more specific drug delivery to tumors [3,4,5], using ligands such as aptamers, polysaccharides, peptides, proteins, and antibodies to generate drug-bioconjugates and functionalized nanoparticles, which selectively recognize membrane receptors of tumor cells to achieve an “active” tumor targeting [3].

Antibody-drug conjugates (ADC) are well-implemented as targeting agents in oncology, with many ADC approved by National Agencies for Drug Regulation for various hematological cancers and solid tumors [6]. More recently, short synthetic peptides have gained interest as targeting bioconjugates for cancer imaging and treatments, due to their high specificity of interaction and excellent biocompatibilty [7,8,9,10,11]. Peptides specifically recognize cell membrane-associated proteins on a subset of cell types and can also exert antiproliferative [12], antiangiogenic [13], and cytotoxic [14,15] activities. Furthermore, several cell-penetrating peptides (CPP) have also been synthesized [9,11], after the demonstration that short domains of the transcription protein trans-activator of the human immunodeficiency virus [16] and Drosophila Antennapedia homeodomain [17] showed the remarkable ability to diffuse across cell membranes.

Nanoparticles (NPs) embedding drugs have appeared as a suitable strategy to prevent the rapid elimination of therapeutic payloads and achieve a prolonged plasma drug concentration [18,19]. This strategy was reinforced by the discovery of the enhanced permeability and retention effect (EPR) in rodents [20], defined as the extravasation of macromolecules and nanovectors, through the fenestrated and leaky blood vessels, as well as their accumulation within tumors [2,21,22]. However, only few nanomedicines have been approved by regulatory authorities and reached the clinical practice [3], because of their limited efficacy and concerns, regarding their toxicity [23,24]. The lack of efficacy could be due, at least in part, to the fact that the EPR effect may not be as predominant in humans, compared to mice, because of differences in hemodynamics between the two species, and tumor heterogeneity in humans. In order to improve tumor targeting, NPs were functionalized with peptide and protein ligands of the membrane receptors expressed in tumor cells [5,25,26,27,28]. While recent articles describe increases in tumor targeting by peptide-decorated NPs [3,5], others conclude to the negligible efficiency because of the reduced diffusion of NPs into solid tumors and their internalization by macrophages within the tumors [29,30], as well as the mononuclear phagocyte system (MPS) in the liver and spleen [2].

Hepatocellular carcinoma (HCC), the most predominant primary liver malignancy worldwide, belongs to the group of refractory cancers resistant to chemotherapies. Only a small fraction of patients are eligible for curative surgical treatments at the early stages of the disease [31]. For most patients, palliative neoadjuvant therapies, such as transarterial chemoembolization and radioembolization [32], and systemic chemotherapies are proposed. In 2007, the FDA approved the tyrosine kinase inhibitor sorafenib as the first line of treatment for advanced HCC [33]. In the last five years, successive phase II and III trials demonstrated the efficiency of several others tyrosine kinase and angiogenesis inhibitors against HCC, including regorafenib [34], lenvatinib [35], cabozantinib [36], and ramucirumab [37,38]. Following these clinical trials, these new drugs were approved as first- or second-line systemic therapy options for patients with unresectable HCC and/or to reduce the risk of recurrence following surgical resection [32,39]. The use of immunotherapy with nivolumab and pembrolizumab were also approved for advanced HCC after tumor progression under sorafenib and lenvatinib [39,40,41,42]. These immunotherapies show, however, moderate improvement on the overall survival and for a limited number of patients [43]. Clinical trials are currently evaluating their use at earlier stages of HCC as neoadjuvant [32] and in association with other therapeutic approaches such as the radioembolization [44]. Nevertheless, HCC remains among the most refractory cancers, and alternative strategies are urgently needed.

While many bioconjugates and NPs have been designed for both imaging and therapy in oncology [45,46,47], only few articles described the evaluation of peptide bioconjugates and peptide-functionalized NPs for the theranostics of HCC and most studies reported rather disappointing results [10,48]. These data emphasize the need to identify specific peptides for designing novel peptide bioconjugates and peptide-decorated NPs targeting hepatic cancer cells more efficiently. Some peptides have been identified mainly by phage display, using hepatoma cells or proteins overexpressed in HCC, such as the glypican-3, as baits [49,50,51,52,53]. These peptides, either covalently bound to fluorescent molecular probes to produce bioconjugates or engrafted onto NPs, exhibit tropism toward hepatoma cells in vitro and in xenograft HCC mouse models. However, these studies rarely investigate the internalization of nanovectors by normal human hepatocytes and macrophages. These studies often use rodent models of HCC, with ectopic implantation of human hepatoma cells in immuno-incompetent nude mice, which lack the active compartment of the MPS, such as macrophages. The development of novel bioconjugates and functionalized NPs promoting the uptake by cancer hepatic cells over the phagocytosis by macrophages would be of interest for the treatment of HCC [54,55].

In this context, our first objective was to identify the peptides that exhibit a strong tropism towards human hepatoma cells, compared to normal hepatocytes in primary culture, by screening a collection of synthetic hepatotropic peptides selected from the literature, but never compared together in the same study. For this task, biotinylated peptides were bound to fluorescent streptavidin to form model bioconjugates, or engrafted through streptavidin bridging onto biotinylated NPs prepared from the amphiphilic block copolymer biotin-poly(ethylene glycol)-*block*-poly(benzyl malate) (biot-PEG-*b*-PMLABe) and encapsulating a hydrophobic fluorescent probe into their hydrophobic PMLABe inner core [56,57,58]. Herein, we demonstrated that the peptides CPB and GBVA10-9 derived from the circumsporozoite protein of *Plasmodium berghei* and George Baker (GB) virus A, respectively, showed the highest tropism towards hepatoma cells. Using these peptides, the second aim of this study was to investigate whether their specific amino acid sequence was the main determinant to drive the cell uptake of peptide bioconjugates and peptide-decorated NPs by hepatoma cells. We generated scrambled and/or mutated control peptides for both CPB and GBVA10-9 and compared the uptake of native versus control peptide bioconjugates and -NPs in various hepatoma cell lines. We demonstrated that CPB and GBVA10-9 native peptide bioconjugates were more efficiently internalized that their control counterparts. The internalization of PEGylated PMLABe-based NPs by hepatoma cells was also strongly enhanced by CPB- and GBVA10-9 peptides through a mechanism that involves, at least in part, the peptide’s binding to apolipoproteins and HDL, which promoted interaction with cell membrane receptors. Then, we set up a coculture system mixing Huh7 human hepatoma cells and primary macrophages in order to determine whether CPB and GBVA10-9 peptide–streptavidin conjugates could favor the uptake by hepatoma cells over that of macrophages in this in vitro model evaluating the cell competition for the internalization of our peptide-streptavidin bioconjugates.

## 2. Materials and Methods

### 2.1. Reagents and Peptides

William’s E medium, RPMI 1640 medium, penicillin/streptomycin (5000 IU/mL, 5000 µg/mL), L-glutamine (2 mM), trypsine-EDTA, Dulbecco’s phosphate-buffered saline (PBS), CellLight^TM^ lysosomes-GFP, streptavidin DyLight^TM^ 488 conjugated, streptavidin DyLight^TM^ 633 conjugated, Pierce^®^ streptavidin agarose resin, Hoechst 33342, and 16% formaldehyde were purchased from Thermo Fisher Scientific (Waltham, MA, USA). Fetal bovine sera (FBS) were from GE Healthcare Life Sciences (Villabe, France), Eurobio Scientific (Les Ulis, France), and Biosera (Grosseron distributor, Coueron, France). Human serum pooled male was obtained from Biotrend Chemikalien GmbH (Köln, Germany). Minimum essential medium (MEM) Eagle, dimethyl sulfoxide (DMSO), genistein, chlorpromazine hydrochloride, insulin from bovine pancreas, biotin, latrunculin A, and saponin were purchased from Sigma-Aldrich (St. Louis, MO, USA). Hydrocortisone hemisuccinate was from Serb (Paris, France). Streptavidin recombinant was purchased from Anaspec (Fremont, CA, USA). Human CD14 microbeads and recombinant human granulocyte macrophage colony-stimulating factor (rhGM-CSF) were purchased from Miltenyi Biotec (Bergisch Gladbach, Germany). The α-biotin-ω-carboxy poly(ethylene glycol) (BiotPEG_n_COOH), with a weight average molar mass (Mw) of 3000 g/mol (*n* = 62), was purchased from Iris Biotech GMBH (Marktredwitz, Germany). DiD Oil (1,1′-dioctadecyl-3,3,3′,3′-tetramethylindodicarbocyanine perchlorate) was purchased from Invitrogen (Thermo Fischer Scientific, Illkrich-Graffenstaden, France). Recombinant human apolipoproteins A1 (ApoA-1) and E3 (ApoE-3) were obtained from Bio-Techne (Noyal-Châtillon-sur-Seiche, France). All peptides (Table 1), modified at the N-terminal end by adding a biotinylated lysine, were synthesized by Eurogentec (Seraing, Belgium), with purity rates of >85%, as determined by high-pressure liquid chromatography. The HDL purification kit was purchased from MyBioSource (CliniSciences, Nanterre, France).

### 2.2. Materials

Characterization of biotinylated-PEG_62_-*b*-PMLABe_73_ copolymer was performed using nuclear magnetic resonance spectra (^1^H NMR) and were recorded on a Bruker Advanced III 400 spectrometer (Bruker, Wissembourg, France). Data are not shown in the manuscript. Weight average molar mass (Mw) and dispersity (Ð = Mw/Mn) values were measured by size exclusion chromatography (SEC) in THF at 40 °C (flow rate = 1.0 mL/min) on a GPC2502 Viscotek apparatus (Malvern Instruments Ltd., Malvern, UK), equipped with a refractive index detector Viscotek VE 3580 RI (Malvern Instruments Ltd., Malvern, UK), guard column Viscotek TGuard org 10 × 4.6 mm, LT5000L mixed medium org 300 × 7.8 mm gel column (Malvern Instruments Ltd., Malvern, UK), and GPC/SEC OmniSEC software (OmniSEC™4.6.1, Malvern Instruments Ltd., Malvern, UK). The polymer samples were dissolved in THF (2 mg/mL). All elution curves were calibrated with poly(styrene) standards. Dynamic light scattering (DLS) and zeta potential (electrophoretic light scattering, ELS) measurements were performed on a Nano-sizer ZS90 (Malvern Instruments Ltd., Malvern, UK) at 25 °C, with a He–Ne laser (Malvern Instruments Ltd., Malvern, UK) at 633 nm and with a detection angle of 90°.

Fluorescent cells were visualized using a Zeiss AxioVert A.1 microscope, coupled with a Colibri.2 illumination system (Carl Zeiss Microscopy GmbH, Oberkochen, Germany). Intracellular localization was evaluated using a Leica SP8 AOBS resonant scanning confocal microscope, with an inverted Leica DMI 6000 microscope stand (Leica Microsystems, Wetzlar, Germany), using the Microscopy Rennes Imaging Center (MRic) facility of the Biology and Health Federative research structure Biosit (SFR Biosit, University of Rennes 1, France). Images were processed using the open-source software ImageJ/Fiji. The fluorescence emitted by cells was quantified by flow cytometry (LSRFortessa™ X-20, Becton Dickinson), using the cytometry core facility of the SFR Biosit (University of Rennes 1, France). Cytometry data were analyzed using the FlowLogic software (7.2.1 version, Inivai Technologies, Mentone Victoria, Australia).

### 2.3. Cell Culture

HepaRG cells were cultured as previously described [59]. Briefly, progenitors HepaRG were cultured in William’s E medium, supplemented with 2 mM l-glutamine, 50 IU/mL penicillin, 50 µg/mL streptomycin, 5 mg/L insulin, 10^−5^ M hydrocortisone hemisuccinate, and 10% FBS. To obtain differentiated HepaRG cells, progenitors were cultured for 14 days to obtain confluent quiescent cells and maintained for two additional weeks in medium supplemented with 2% DMSO. MEM medium, supplemented with 10% FBS, 50 IU/mL penicillin, 50 μg/mL streptomycin, and 2 mM L-glutamine, was used to culture the HepG2 and Huh7 cells. Human primary hepatocytes, obtained from the Centre de Ressources Biologiques (CRB) of Rennes, were cultured in William’s E medium, supplemented with 2 mM L-glutamine, 50 IU/mL penicillin, 50 µg/mL streptomycin, 5 mg/L insulin, 10^−5^ M hydrocortisone hemisuccinate, 10% FBS, and 2% DMSO. The bronchial carcinoid NCI-H727 cells were cultured in RPMI 1640 medium containing 10% FBS, 50 IU/mL penicillin, 50 μg/mL streptomycin, and 2 mM L-glutamine.

Human peripheral blood mononuclear cells were isolated from the buffy coat of healthy donors (Etablissement Français du Sang, Rennes, France) by centrifugation on UNI-SEP maxi U10 (Novamed). Monocytes were isolated using anti-human CD14 antibodies conjugated magnetic MicroBeads (Miltenyi Biotec SAS, Paris, France) and were plated at a density of 0.5 × 10^5^ cells per well in 48-well plates. HPM were obtained after differentiation of monocytes with 50 ng/mL rhGM-CSF in RPMI 1640 medium, supplemented with 5 IU/mL penicillin, 5 mg/mL streptomycin, 2 mM L-glutamine, and 10% FBS during 7 days, as previously described [60]. After 7 days of differentiation, the culture medium was removed, and 1 × 10^5^ cells Huh7 hepatoma cells expressing the green fluorescent protein (GFP) were added in HPM’s wells. RPMI 1640 medium, supplemented with 10% FBS, 50 IU/mL penicillin, 50 μg/mL streptomycin, and 2 mM L-glutamine, was used for the coculture. GFP-expressing Huh7 cells were produced by lentiviral transduction of Huh7 cells plated at low cell density (10^5^ cells per well in 24-well plates), with pre-made lentiviral particles (ILV-EF1-GFP) obtained from Flash Therapeutics (Toulouse, France). All cells were incubated at 37 °C, with 5% humidified CO_2_.

### 2.4. Preparation of Peptide-Functionalized PMLABe-Based Nanoparticles

Nanoparticles (NPs) embedding the fluorescence probe DiD Oil were prepared by nanoprecipitation starting from the biotinylated-PEG_62_-*b*-PMLABe_73_ (Biot-PEG_62_-*b*-PMLABe_73_) amphiphilic block copolymer, which was synthesized as previously described, with slight modifications [56]. Briefly, the Biot-PEG_62_-*b*-PMLABe_73_ was obtained by anionic ring-opening polymerization (aROP) of benzyl malolactonate (MLABe), in the presence of the tetraethylammonium salt of the α-biotin-ω-carboxylate-PEG_62_ as an initiator, which was previously obtained by mixing one equivalent of ω-biotin, α-carboxylic acid-PEG_62_, and one equivalent of tetraethylammonium hydroxide. The molar mass of the PMLABe block was fixed by the ratio monomer (MLABe)/initiator at 15,000 g/mol. After consumption of all the monomer, the crude amphiphilic block copolymer was purified by precipitation in an excess of ethanol. After drying under vacuum, the purified Biot-PEG_62_-*b*-PMLABe_73_ was analyzed by ^1^H NMR (structure and molar mass of PMLABe block) and SEC (average molar mass and dispersity).

^1^H NMR (400.1 MHz; (CD_3_)_2_CO), δ (ppm): 2.83 (s, 2nH, CO_2_CH_2_C_6_H_5_), 3.56–3.59 (m, 4 mH (m = 62), Biot (CH_2_CH_2_O)_62_), 5.00–5.05 (m, 2nH, CHCH_2_CO_2_), 5.41–5.44 (m, 1nH, CHCH_2_CO_2_), and 7.19–7.23 (m, 5nH, CO_2_CH_2_C_6_H_5_).

M_NMR_ = 15,500 g/mol for the PMLABe block – M_block copolymer_ = 17,900 g/mol.

SEC (THF, polystyrene standards, 1 mL/min): Mw = 8200 g/mol, Ð = 1.20.

NPs encapsulating the fluorescence probe DiD Oil were prepared by the nanoprecipitation method previously described [61]. Five mg of Biot-PEG_62_-*b*-PMLABe_73_ were dissolved into 100 µL of DMF. Fifty µL of a DiD Oil solution, at a concentration of 1 mg/mL in DMF, were added into the polymer solution (1 wt% the mass of the block copolymer, i.e., 0.5 mg of DiD Oil). This mix was nanoprecipitated into 1 mL of distilled water, under vigorous stirring, for 1 h at room temperature. Free DiD Oil was removed by filtration through a Sephadex PD10 column, while DiD Oil-loaded Biot-PEG_62_-*b*-PMLABe_73_-based NPs eluted in 3.5 mL of water. All the NP’s suspensions (~2.85.10^12^ NPs/mL) were characterized by DLS and ELS: hydrodynamic diameter = 89 ± 10 nm, dispersity index (PDI) = 0.25 ± 0.1, and zeta potential = −40 ± 5 mV.

The engrafting of the biotinylated peptides onto Biot-PEG_62_-*b*-PMLABe_73_ NPs was obtained via a two-step streptavidin/biotin coupling [57]. For one well of 24-well plates, 3 μL of recombinant streptavidin (stock solution at 182 μM) and 7 μL of biotinylated peptides (stock solution 150 μM) were mixed with 5 μL of PBS and incubated together for 1 h at 4 °C. Then, 5 μL of NPs (5 mg of copolymer/3.5 mL) were added and incubated with the streptavidin and biotinylated peptide complexes for another 1 h at 4 °C. After dilution of the peptide-NP mixes (20 μL) in culture medium (500 μL/well of 24-well plates), the final concentrations of streptavidin, peptide, and copolymer were 1 µM, 2 µM, and 1 µM, respectively. Control NPs were generated by coupling PEGylated biotin, instead of peptides, onto NPs at the same final concentration. To measure the size and zeta potential of NPs during incubation, suspensions of control and peptide-functionalized NPs (80 µL) were diluted in 1720 µL of culture medium and analyzed by DLS and ELS. Then, 200 µL of FBS were added in batches of peptide-NPs, which were incubated for 24 h at 37 °C, in absence of cells, before analysis by DLS and ELS.

### 2.5. In Vitro Cellular Uptake of Peptide-Conjugates and NPs

The progenitor HepaRG, HepG2, Huh7, and NCI-H727 cells were seeded in 24- or 48-well plates at 6.5 × 10^4^, 2 × 10^5^, 6.5 × 10^4^, and 1.5 × 10^5^ cells per cm^2^ (~80% confluency), respectively, and incubated with peptide conjugates and NPs one day after seeding. Differentiated HepaRG cells were obtained after 30 days of culture (Section 2.3), prior to the uptake assay. Human primary hepatocytes were plated at 2 × 10^5^ cells/cm^2^ and treated 2 to 3 days after seeding. For the coculture between HPM- and GFP-expressing Huh7, cells were incubated 1 day after setting up the coculture.

Biotin and biotinylated peptides were conjugated to DyLight^TM^ 488 or 633 fluorescent streptavidin for 1 h at 4 °C, prior to incubation with cells. Then, streptavidin complexes were added to 180 µL of appropriate culture medium at final concentrations of 0.4 µM streptavidin and 1.2 µM of biotin or biotinylated peptides. The incubation was carried for 24 h. For the experiments of cell uptake inhibition, cells were incubated with peptide–streptavidin conjugates for 4 h, in the absence or presence of genistein (100 μM) or latrunculin A (500 nM), to avoid degradation of the compounds or cytotoxicity.

Control and peptide-functionalized NPs were prepared as described in Section 2.4. Culture medium was withdrawn and replaced by 500 µL (in 24-well plates) of fresh culture media, containing NPs at a final concentration of 25 μg/mL in copolymer, corresponding to ~5.10^10^ NPs/mL. For NP’s uptake inhibition, cells were incubated with NPs for 4 h, in the absence or presence of both chlorpromazine (20 μM) and genistein (100 μM), in culture medium. To evaluate the effects of HDL and ApoE3 on internalization of NPs in HepaRG cells, HDL were purified from 10 mL of human serum, according to the manufacturer’s instructions (MyBioSource), and recombinant human ApoE3 were purchased. Purified HDL and ApoE3 were added in various amounts, ranging from 0.1 to 2 μL and 0.1 to 8 ug/mL, respectively, and mixed to control and peptide-functionalized NPs, prior to the dilution in culture media and incubation with cells.

After incubation at 37 °C in a humidified atmosphere of 5% CO_2_, cell monolayers were washed twice with PBS and photographs were acquired using fluorescence microscope. Then, the cells were detached with trypsin-EDTA and resuspended in complete medium for flow cytometry analysis. Dot plots of forward scatter (FSC: x axis) and side scatter (SSC: y axis) allowed us to gate the viable single cells (Appendix A). Untreated cells were used to determine autofluorescence, arbitrary set at ~30 for all cell types. The fluorescence emitted by streptavidin DyLight^TM^ 488 was measured using the FITC-A channel, and the fluorescence emitted by both streptavidin DyLight^TM^ 633 and NPs encapsulating DiD Oil were detected using the APC-A channel. Means of fluorescence were expressed as fluorescence intensity (MFI) of the single cell population (Appendix A). The effects of peptides on the cell uptake of DL488-SA-peptide bioconjugates and peptide-decorated NPs was evaluated with the ratio of MFI for cells incubated with peptide-functionalized streptavidin or NPs versus MFI for cells incubated with streptavidin or NPs without peptides.

For confocal microscopy, HepaRG cells grown in 8-well coverglass Nunc^TM^ Lab-Tek^TM^ (Thermofisher scientific) were incubated with peptides functionalized-NPs at 37 °C for 24 h, fixed using 4% paraformaldehyde in PBS at room temperature (RT) for 10 min, and permeabilized and saturated with 0.5% saponin and 3% FBS in PBS for 1 h at RT. Then, cells were incubated with rabbit anti-Rab7 monoclonal antibody (D95F2, Cell Signaling Technology, Danvers, MA, USA) overnight at 4 °C. Cells were washed three times with PBS and incubated with fluorescein isothiocyanate (FITC)-conjugated anti-rabbit IgG antibody (Eurobio Scientific, Les Ulis, France) for 1 h at room temperature, protected from light. Nuclei were stained with Hoechst for 30 min at RT in the dark. The images were acquired using SP8 confocal microscope with LAS AF software and analyzed with Fiji/ImageJ.

### 2.6. Opsonization of NPs and Peptide-Streptavidin Agarose Resin

The opsonization of NPs was studied using a protein adsorption assay followed by western blot analysis. Biot-PEG_62_-*b*-PMLABe_73_-based NPs (functionalized or not) were incubated in William’s E medium supplemented with 10% human serum during 24 h at 37 °C under 5% CO_2_ humidified atmosphere. Adsorption of plasma proteins onto peptide-streptavidin agarose resin (Pierce^®^ Thermofisher Scientific, USA) was studied by binding peptides (2 μg, 5 μL) onto 10 μL of settled of resin, followed by incubation for 30 min at 4 °C with 10% human serum (25 μL) and PBS (200 μL).

NPs were collected by centrifugation at 14,000× *g* for 30 min, while agarose beads were spun down at 5000× *g* for 1 min at 4 °C. The pellets were washed once with cold PBS (500 µL), prior to denaturation of micelles and bound proteins with 50 µL of denaturating buffer (Tris-HCl 100 mM, pH 6.8, bromophenol blue 0.2%, sodium dodecyl sulfate 8%, glycerol 20%, and β-mercaptoethanol 5%) and 50 µL of MOPS/SDS buffer pH 7.7 (Thermofisher Scientific, Waltham, MA, USA). Samples were boiled in a water bath for 10 min. Proteins were loaded and separated by electrophoresis on polyacrylamide gels (iD PAGE gel, 4–12%, Eurogentec, (Seraing, Belgium) and then transferred to nitrocellulose membranes (iBlot^®^ 2NC Mini Stacks, Thermofisher Scientific, Waltham, MA, USA). Standard PageRuler^TM^ Plus prestained protein ladder (Thermofisher Scientific, Waltham, MA, USA) was run in parallel. The membranes were blocked with 3% bovine serum albumin (BSA) fraction V (Eurobio) in 1X Tris-buffered saline, 0.1% Tween 20 (TBST) at room temperature (RT), for 1 h, then incubated overnight at 4 °C, with the following primary antibodies diluted in TBST containing 3% BSA, followed by rinsing three times with TBST: mouse anti-human complement C3 proteins (B-9), mouse anti-human apolipoprotein C-I (Y-13) and mouse anti-human apolipoprotein-E (A1.4) (purchased from Santa Cruz Biotechnologies (Dallas, TX, USA); goat anti-human transferrin (1205Y2) and goat anti-human albumin (1140V7) (obtained from Kent laboratories (Bellingham, WA, USA); and goat anti-human IgG (A-0293) (from Sigma-Aldrich (St. Louis, MO, USA)). The membranes were then incubated for 1 h at RT with the appropriate horseradish peroxidase (HRP) secondary antibodies: polyclonal rabbit anti-mouse, polyclonal goat anti-rabbit, and polyclonal rabbit anti-goat (Dako, Glostrup Kommune, Denmark) After incubation, the membranes were washed three times with TBST and developed using the SuperSignal^TM^ WestDura chemiluminescent substrate kit for HRP detection (Thermofisher Scientific, Waltham, MA, USA), according to the manufacturer’s instructions. The proteins were visualized with the Fusion FX system (Vilber-Lourmat, Eberhardzell, Germany).

### 2.7. Statistical Analysis

All quantitative data were expressed as the mean ± standard deviation (SD). Statistical analyses were performed using GraphPad Prism version 5.0 (GraphPad Software, San Diego, CA, USA). Differences between two groups were analyzed using two-tailed Mann–Whitney *U* test. A non-parametric Kruskal–Wallis test with Dunns’ post-test was used to compare the means of more than two groups. Significant differences are presented as * *p* < 0.05, ** *p* < 0.01, *** *p* < 0.001, and ^ns^ not significant.

## 3. Results

### 3.1. Selection of Hepatotropic and Non-Hepatic Peptides

We selected, from the literature, 11 peptides previously reported to show hepatotropism towards in vitro and/or in vivo models of human and murine hepatic cancer cells as baits (Table 1). RGD peptide has been widely used to target many different types of cancer cells [62], including hepatoma cells [55,57], via its well-documented affinity for the α_v_β_3_ integrins. We selected a cyclic RGD-like peptide, which constituted our reference in the experiments of cell uptake, as well as its RAD negative control. We also used the peptides L5-2 [63] and SRIF-14 [10], derived from the functional domains of natural ligands of the glypican 3 protein and somatostatin receptors, respectively, which are frequently overexpressed in HCC. We selected the short peptides HCBP1 and HCBP4 [64], A54 [65], DKN [66], CGK [67], and SP94 [68], which were previously identified by phage-display using hepatoma cells, although their cellular targets remain unknown.

**Table 1 pharmaceutics-14-00804-t001:** Peptides and negative control peptides. Eleven peptides were selected to study their tropism towards human hepatoma cells in vitro: RGD, L5-2, SRIF-14, HCBP1 and HCBP4, A54, DKN, CGK, SP94, CPB, and GBVA10-9. The peptides CP15 and ATW, initially discovered for their binding to human colon cancer and endothelial cells, respectively, were selected as controls of non-hepatotropic peptides. RAD, CPBscr, GBVA10-13, and GBVA10-9scr peptides are mutated or scrambled peptides used as negative control for the in vitro cell uptake assays. The amino acid sequences, cellular and/or molecular targets, length, and molecular weight are indicated. References describing the cellular/molecular tropism of these peptides are indicated.

Name	Sequence	Target (Receptor/Cells or Tissue)	Length	MW (g/mol)	Ref.
RGD	Cyclo(RGDYK)(Biotin)	A_v_β_3_ integrins	5	846	[62]
L5-2	YFLTTRQK(Biotin)	Glypican-3	8	1282	[63]
SRIF-14	AGC(S-)KNFFWKTFTSC(S-)K(Biotin)	Somatostatin receptor	15	1991	[10]
HCBP1	FQHPSFIK(Biotin)	ND/HepG2	8	1230	[64]
HCBP4	PLPTLPLK(Biotin)	ND/HepG2	8	1103	[64]
A54	AGKGTPSLETTPK(Biotin)	ND/BEL-7402	13	1512	[65]
DKN	DKNLQLHK(Biotin)	ND/Rat liver	8	1220	[66]
CGK	CGKRK(Biotin)	Protein p32/gC1qR/HABP1/Tumor stroma and angiogenic endothelial cells	5	816	[67]
SP94	SFSIIHTPILPLK(Biotin)	ND/Malhavu (+ 59T, Hep3B, HepG2, NTUBL, SKHep1)	13	1691	[68]
CPB	Acetyl-CKNEKKNKIERNNKLKQPPK(Biotin)	Heparan sulfate proteoglycan	20	2705	[69]
GBVA10-9	CWVRLGRYLLRRLKTLFTK(Biotin)	ND. Anti-HCV activity	19	2649	[70]
CP15	VHLGYATK(Biotin)	ND/HT29 and SW480	8	1113	[71]
ATW	ATWLPVPK(Biotin)	Neuropilin-1/Endothelial cells	8	1137	[72]
RAD	Cyclo(RADYK)(Biotin)	Negative control of RGD	5	860	[5]
CPBscr	EKIPNKEKPNNKNKERQKK(Biotin)	Negative control for CPB	19	2577	None
GBVA10-13	KWVRLGRKLLRRLKKPFKK(Biotin)	Negative control for GBVA10-9	19	2677	[70]
GBVA10-9scr	LTCLKGRLVRWRLRTYFLK(Biotin)	Negative control for GBVA10-9	19	2649	None

We included two additional peptides, CPB and GBVA10-9, derived from proteins expressed in pathogens infecting hepatocytes. The 19 amino-acid CPB peptide from the circumsporozoite protein (CSP) of *Plasmodium berghei* were involved in the infection of hepatocytes by the parasite and previously used to prepare CPB-containing liposomes that target mouse liver [69]. The CPB peptide contains a consensus heparan sulfate proteoglycan binding sequence. We have also chosen the amphipathic helical GBVA10-9 peptide derived from the NS5 protein region of selected *Flaviviridae* viruses because of its potent anti-hepatitis C virus activities [70]. The heptapeptide peptides CP15 [71] and ATW [72], initially discovered for their binding to human colon cancer and endothelial cells, respectively, were selected as controls of non-hepatotropic peptides in the screening. While the CP15 cell target is unknown, the ATW peptide binds to neuropilin-1, a vascular endothelial growth factor (VEGF) co-receptor [72].

### 3.2. Cell Uptake of Streptavidin-Peptide Bioconjugates

The biotinylated peptides were bound to fluorescent DyLight^TM^ 488 streptavidin (DL488-SA), at a 3:1 molar ratio, to form model bioconjugates [73] and evaluate the ability of the selected peptides to trigger the uptake of streptavidin in HepaRG, Huh7, and HepG2 hepatoma cells, human hepatocytes in primary culture, and non-hepatic NCI-H727 bronchial cancer cells (Figure 1). In a first experiment, we determined whether the control biotin–streptavidin conjugate without peptides (Figure 1A) was internalized by these different cell types, as well as to evidence a putative difference in “passive” cell uptake between cell types (Figure 1B). The mean of fluorescence intensity (MFI), analyzed by flow cytometry, in untreated cells (auto-fluorescence) was arbitrarily set at ≈30 units (a.u) for all cell types. The MFI measured after 24 h of incubation with the control biotin–streptavidin conjugate revealed a significant disparity in the binding and uptake of this control bioconjugate between the cell types (Figure 1B). We observed a very low fluorescence for primary human hepatocytes, an intermediate MFI for HepG2, Huh7, and NCI-H727 cells, as well as a much higher fluorescence for the progenitor and differentiated HepaRG cells. Nevertheless, this overall binding/uptake of the control biotin–streptavidin complex remained low as demonstrated by the weak fluorescence signal observed by flow cytometry (Figure 1B) and microscopy in HepaRG cells (Figure 1C).

In order to assess the influence of the selected hepatotropic and non-hepatotropic peptides on the cell uptake of streptavidin (Figure 1D–F), we expressed the results as the ratio of MFI between cells incubated with DL488-SA-peptide bioconjugates versus cells incubated with the control biotin–streptavidin bioconjugate, in order to consider the differences in cell uptake of the control complex observed with the different cell types (Figure 1B,C). Among hepatotropic peptides, only the CPB, GBVA10-9, and SRIF-14 peptides significantly enhanced the uptake of bioconjugates in hepatoma cells, compared to that obtained with the control biotin–streptavidin complex (Figure 1D). The strong increase in the uptake of GBVA10-9-DL488-SA complex was quantified by flow cytometry and confirmed by fluorescence microscopy in HepaRG cells, compared to that observed with the control biotin-DL488-SA conjugate (Figure 1C and Appendix A). The cell uptake of CPB- and GBVA10-9-DL488-SA complexes occurred mainly through endocytosis, as demonstrated by the intracellular accumulation of peptide-DL488-SA complexes, evidenced by confocal microscopy (Appendix A), which was strongly decreased by endocytosis inhibitors chlorpromazine and genistein.

The increase in bioconjugate’s uptake was particularly marked in Huh7 cells, with MFI that were 28 and 568 times greater in the presence of CPB and GBVA10-9 peptides, respectively, compared to those detected with the control bioconjugate. In human hepatocytes, only GBVA10-9 and SRIF-14 peptides significantly increased the bioconjugates’ uptake (Figure 1E), but to a much weaker extent, compared to hepatoma cells, since the MFI ratios measured in presence of GBVA10-9-DL488-SA bioconjugate was only 16.8 times greater than that measured with the control biotin-DL488-SA bioconjugate (Figure 1E). Similarly, the CPB, GBVA10-9, and SRIF-14 peptides also enhanced streptavidin’s uptake by bronchial cancer NCI-H727 cells, since the ratios between MFI in cells exposed to peptide- versus biotin-DL488-SA complexes for these three peptides were 5.5, 61.4, and 3, respectively (Figure 1F). Unexpectedly, the SP94, which weakly affected the streptavidin uptake in hepatoma cells, induced a 16-fold increase in MFI of the NCI-H727 cells.

Together, our data demonstrated that, among the peptides selected for this screen, based upon previous results reporting their binding to hepatoma cells, only the CPB and GBVA10-9 peptides showed a high tropism towards human hepatoma cells in our experimental conditions.

To assess the role of the specific amino acid sequence of the CPB and GBVA10-9 peptides in the bioconjugate’s uptake by hepatoma cells, we synthesized three control peptides, i.e., the CPBscr, GBVA10-9scr, and GBVA10-13 peptides (Table 1). The scrambled CPBscr and GBVA10-9scr peptides contain the same amino acid composition, compared to their native CPB and GBVA10-9 counterparts, but with a random sequence. The GBVA10-13 control was derived from the GBVA10-9 peptide whose modifications, made to the amino acid sequence, resulted in loss of anti-HCV activity of the native GBVA10-9 [70]. The peptide-DL488-SA bioconjugates were incubated with hepatoma cells and NCI-H727 cells to determine whether modifications of the amino acid sequence within the control peptides affected the cell uptake of bioconjugates. The relative MFI were measured by flow cytometry in untreated cells (control) and in cells incubated with peptide-DL488-SA bioconjugates, which allowed a direct comparison of the overall uptake in the different cell lines (Figure 2).

The results first evidenced that the uptake of native GBVA10-9-DL488-SA bioconjugate was much higher than that measured with the CPB-DL488-SA complex, since 20- to 60-fold differences in fluorescence levels were observed between the two bioconjugates. The data also confirmed that CPB and GBVA10-9 peptides strongly enhanced the uptake of bioconjugates, compared to that of the control biotin-DL488-SA complex for all cell types (Figure 2). Moreover, the uptake was higher in hepatoma cells than in NCI-H727 bronchial cells, except for differentiated HepaRG cells incubated with the CPB-DL488-SA bioconjugate (Figure 2A,B). Importantly, cells incubated with the CPBscr-DL488-SA complex exhibited MFI almost equivalent to those incubated with peptide-free control bioconjugate, which indicated that the modifications of the amino acid sequence of the CPB peptide induced a complete loss of cell recognition (Figure 2A,C). Similarly, the MFI measured for cells exposed to GBVA10-9scr- and GBVA10-13-DL488-SA bioconjugates were strongly reduced, compared to those found with GBVA10-9-DL488-SA complex, although they remained significantly higher than those of biotin-DL488-SA controls (Figure 2B,C).

These data strongly suggested that the specific amino acid sequences of both CPB and GBVA10-9 peptides were the major determinant of the peptide-mediated uptake of the fluorescent streptavidin bioconjugate by hepatoma cells.

### 3.3. Cell Uptake of Peptide-Decorated Nanoparticles

We next investigated whether the hepatotropic peptides selected (Table 1) could enhance the uptake of nanocarriers by hepatoma cells, in order to compare the hepatotropism between peptide-NPs and peptide-DL488-SA bioconjugates. The biotinylated peptides were bound to nanoparticles (NPs), prepared from biotin-poly(ethylene glycol)-*block*-poly(benzyl malate) (Biot-PEG_62_-*b*-PMLABe_73_) harboring biotin at their surface [74,75,76] and containing the hydrophobic DiD Oil fluorescent probe (Figure 3A), which allowed the quantification of NP’s cell uptake by flow cytometry and microscopy [57].

In a first experiment, we compared the uptake of control biotin–streptavidin-biotin–PEG_62_-*b*-PMLABe_73_ NPs without peptide(s) (Figure 3A) in HepaRG, Huh7, and HepG2 hepatoma cells, human hepatocytes in primary culture, and NCI-H727 cells (Figure 3B). As previously observed with the control biotin–streptavidin bioconjugates (Figure 1B), we found important differences in the MFI of the different cell types incubated for 24 h with the biotin–streptavidin biotin–PEG_62_-*b*-PMLABe_73_ NPs, which revealed various levels of NPs’ uptake activities between the cell types (Figure 3B). The human hepatocytes and HepG2 cells internalized low amounts of NPs, Huh7, and NCI-H727 cells showed intermediate MFI, while the progenitor and differentiated HepaRG cells exhibited much higher fluorescence levels.

In order to assess the peptide’s influence on the cell uptake of peptides-NPs (Figure 3D–F), we expressed the results of flow cytometry as the ratio of MFI between cells incubated with peptide-decorated NPs versus cells exposed to control biotin–streptavidin biotin–PEG_62_-*b*-PMLABe_73_ NPs without peptides, in order to consider the differences in the “passive” uptake observed between the cell types (Figure 3B). All the hepatotropic peptides strongly enhanced the NPs’ uptake in the hepatoma cells (Figure 3D). Although the addition of peptides onto NPs significantly increased the NP’s internalization in human hepatocytes (Figure 3E) and NCI-H727 cells (Figure 3F), the positive effects on the cell uptake remained much weaker in these two cell types, compared to that observed in hepatoma cells. The CPB and GBVA10-9 peptides induced the highest increase in NP’s uptake in the hepatoma cells, particularly in HepaRG cells (Figure 3D). The strong accumulation of GBVA10-9-PEG_62_-*b*-PMLABe_73_ NPs in HepaRG cells was confirmed by fluorescence microscopy in HepaRG cells (Figure 3C). Unexpectedly, the non-hepatotropic peptides ATW and RAD also enhanced the NP’s uptake in differentiated HepaRG, Huh7, and NCI-H727 cells, but not in progenitor HepaRG and HepG2 cells (Figure 3D,F).

Our data demonstrated that the addition of peptides onto NPs significantly enhanced their cell uptake. These results also questioned the specificity of the NP-cell interaction mediated by the peptides via their binding to membrane receptors in a sequence-specific manner, since nearly all the peptides strongly increased the NP’s cell uptake, including non-hepatotropic ones in HepaRG, Huh7, and NCI-H727, in contrast with the lack of targeting effect observed with most peptide-DL488-SA bioconjugates.

The role of the amino acid sequence of CPB and GBVA10-9 peptides in the NP’s uptake by hepatoma and NCI-H727 cells was addressed using CPBscr, GBVA10-9scr, and GBVA10-13 control peptides. We measured the relative MFI of cells incubated with NPs harboring CPB and GBVA10-9 peptides, as well as their negative control counterparts, to compare the overall NP’s uptake between cell types and the effects of the mutated control peptides on the NP’s internalization (Figure 4A–C). Our data showed that the MFI were much higher in hepatoma cells than in NCI-H727 cells, confirming the greater uptake of NPs by hepatic cells. We also found that addition of CPBscr onto NPs significantly reduced the cell uptake, compared to that observed with CPB-decorated NPs, except in the progenitor HepaRG cells (Figure 4A,C), although the MFI obtained for cells incubated with CPBscr-decorated NPs remained much higher than those measured for cells exposed to control NPs without peptides. Conversely, the MFI of the HepaRG, Huh7, and NCI-H727 cells incubated with NPs harboring GBVA10-9scr and GBVA10-13 control peptides were either not significantly different or even higher from the MFI measured in cells exposed to GBVA10-9-decorated NPs. In contrast, the NP’s uptake was decreased in HepG2 cells incubated with NPs functionalized with GBVA10-9scr and GBVA10-13 control peptides, compared to the internalization obtained with GBVA10-9-decorated NPs.

These data strongly suggested that the specific amino acid sequence of the CPB peptide contributed, at least for some cell types, to the NP-cell interaction and the cell uptake, since the addition of negative control peptide reduced the NP’s uptake. Our results also indicated that the internalization of GBVA10-9-decorated NPs did not strictly rely on the recognition of binding sites on hepatoma cells mediated by the specific sequence of GBVA10-9, since the two control peptides had no effect or enhanced the NP’s cell uptake peptides.

### 3.4. High Density Lipoprotein (HDL) and Apolipoprotein E Play a Pivotal Role in the Peptide-Decorated NP’s Endocytosis by Hepatoma Cells

To further investigate the mechanism of the NP’s internalization, we used inhibitors of endocytic pathways known to be involved in the uptake of most nanocarriers [5]. HepaRG cells were incubated with NPs harboring the CPB, GBVA10-9, and their control peptides, in the absence or presence of a mix of chlorpromazine and genistein, known to inhibit clathrin- and caveola-mediated endocytic mechanisms [77]. As expected, the treatment with endocytosis inhibitors significantly reduced the MFI of HepaRG cells incubated with NPs functionalized with all the peptides (Figure 5A), confirming that their internalization occurred via endocytosis and probably by macropinocytosis. This was not further investigated in this study. We also studied the intracellular localization of NPs by confocal microscopy, which showed the strong cytoplasmic accumulation of fluorescent nanocarriers containing DiD Oil around cell nuclei, particularly for GBVA10-9- and GBVA10-9scr-decorated NPs (Figure 5B). The DiD Oil staining partially co-localized with the Rab7 GTPase (Figure 5B and Appendix A) known to be associated with membranes of late endosomes. The DiD Oil-containing NPs were also detected within lysosomes expressing the lysosome-associated membrane protein 1 (LAMP1) green, fluorescent fusion protein by confocal microscopy (Appendix A).

After systemic administration, like any other antigen, NPs are recognized by antibodies, proteins of the complement system, as well as many plasma proteins via non-specific interactions, to form a protein corona via the so-called opsonization process [78,79,80]. The composition of the protein corona on NPs is qualitatively and quantitatively dynamic, varies with the physicochemical parameters at the surface of NPs, and partially governs their systemic lifetime and cell uptake [80]. In a previous work, we showed that presence of fetal calf serum (FCS) in the culture medium also leads to opsonization of polymeric NPs affecting cell internalization [81]. To determine whether the nature of the peptides used to functionalize NPs could affect the composition of the protein corona, we studied by immunoblotting the presence of opsonins onto NPs exhibiting CPB, GBVA10-9, and their negative control peptides at their surface (Figure 5C). As expected, we detected albumin and transferrin, two very abundant plasma proteins known to bind to polymeric NPs [80,81]. Their relative levels did not significantly vary between the different peptide-NPs. In contrast, we evidenced quantitative variations for the iC3b protein, apolipoproteins (ApoA-I, ApoC-I, and ApoE), and immunoglobulins (Figure 5C). Notably, ApoA-I and ApoE were much more abundant on NPs harboring GBVA10-9 and its control peptides and, to a lesser extent, on CPB-functionalized NPs, compared their relative levels on the other control NPs.

The amphipathic helical GBVA10-9 peptide was derived from the C5A peptidic domain of the NS5A protein of *Flaviviridae* GB virus A (GBV-A) and showed potent anti-HCV activity by inhibiting the viral entry [70,82]. Interestingly, some amphipathic peptides, such as C5A and GBVA10-9, have α-helical conformations [83] and show antiviral properties [84]. HCV entry into hepatocytes is a complex and multistep process involving viral attachment, receptor binding, and endocytosis, in which the ApoE was shown to play a key role via its specific interaction with the C-terminal α-helix domain of NS5A [85,86]. HCV particles bind to several lipoproteins and transiently interact with low-density lipoprotein (LDL) and high-density lipoprotein (HDL) particles to form lipoviral particles that attach to hepatocytes via the binding to the lipoprotein receptors [87]. These data combined to our results, showing that higher amounts of ApoA-I and ApoE were bound to NPs harboring GBVA10-9 and its control peptides, which prompted us to investigate if apoliproteins could interact with CPB, GBVA10-9, and control peptides.

We set up a pulldown assay by mixing biotinylated CPB, GBVA10-9, and control peptides with human serum, then, the peptides were immobilized onto streptavidin-agarose beads and proteins bound to the beads was analyzed by immunoblotting (Figure 5D). The pulldown assay showed that ApoA-I was detected only on GBVA10-9 agarose beads, while ApoC-I was barely detectable in any samples. In addition, higher amounts of ApoE, iC3b protein, transferrin, and immunoglobulins were found associated with CPB-, GBVA10-9-, and GBVA10-9scr-streptavidin-agarose beads. Albumin was detected in all samples at slightly higher amounts on GBVA10-9 and GBVA10-9scr-streptavidin-agarose beads.

These data confirmed that functionalization of PEG_62_-*b*-PMLABe_73_-based NPs with peptides affected the formation of the protein corona and suggested that apolipoproteins ApoA-I and ApoE, two main components of HDL, may interact with CPB and GBVA10-9 and could regulate the cell uptake of functionalized NPs in hepatoma cells. This hypothesis is further supported by the observation that the hydrodynamic diameter (Dh) of CPB- and GBVA10-9 functionalized NPs is much higher than that of NPs without peptides (Appendix A). In addition, the Dh increased with time during incubation in culture medium supplemented with FCS (Appendix A), suggesting formation of macromolecular aggregates between peptide-functionalized NPs, plasma proteins, and HDL.

In order to study the influence of HDL and apolipoproteins on cell internalization of CPB- and GBVA10-9-functionalized NPs, we incubated HepaRG cells with fluorescent control biotin–streptavidin biotin–PEG_62_-*b*-PMLABe_73_ NPs (without peptide), CPB-, and GBVA10-9-streptavidin–biotin-PEG_62_-*b*-PMLABe_73_ NPs, in the absence or presence of increasing amounts of purified HDL (Figure 5E and Appendix A) and recombinant human ApoE3 protein (Figure 5F), and we analyzed the MFI of cells by flow cytometry. While the overall MFI of the cells incubated with control NPs without peptide was not affected by the addition of HDL, the MFI of HepaRG cells incubated with CPB- and GBVA10-9-functionalized NPs was strongly decreased by HDL in a dose-dependent manner (Figure 5E), suggesting a competition between NPs and HDL for cell uptake. The ApoE3 protein showed opposite effects on cell uptake of CPB- and GBVA10-9-functionalized NPs. While the ApoE3 decreased the internalization of CPB-functionalized NPs, it increased the cell uptake of GBVA10-9-functionalized NPs in the presence of high concentrations of ApoE3 in culture media, compared to the uptake of control NPs (Figure 5F).

Together, these data demonstrated that the internalization of PEG_62_-*b*-PMLABe_73_-based NPs by hepatoma cells was strongly enhanced by the engrafting of CPB- and GBVA10-9 peptides onto NPs, through a mechanism that involves, at least in part, the binding of apolipoproteins and HDL onto these two peptides, which could promote the interaction of NPs with cell membrane receptors.

### 3.5. CPB and GBVA-Derived Peptide–Streptavidin Conjugates Improve Endocytosis by Hepatoma Cells over the Uptake by Macrophages in a Coculture In Vitro Model

We next used CPB and GBVA10-9 peptide–streptavidin bioconjugates (DL633-SA-peptide), which both showed amino acid sequence-dependent tropism towards hepatoma cells (Figure 2) to determine whether these bioconjugates increased the uptake in vitro by hepatoma cells over that of macrophages. We set up a coculture system, associating Huh7 human hepatoma cells and human macrophages (HPM), in order to evaluate the cell competition for the internalization of DL633-SA-peptide complexes. To discriminate the two cell types by flow cytometry, we used Huh7 cells that stably express GFP proteins (Figure 6A). Cocultures were exposed to peptide-DL633-SA bioconjugates and the red fluorescence emitted by the streptavidin was measured in GFP positive Huh7 cells and GFP negative HPM (Figure 6A,B).

In cocultures incubated with DL633-SA-Biot, the MFI was much higher in HPM than Huh7 cells with a 22.7 MFI ratio (Figure 6B–D), confirming that HPM in this coculture model exhibited strong phagocytosis activities of macromolecular complexes. The MFI measured for the native CPB-DL633-SA complex indicated that CPB peptide slightly increased the complex uptake by HPM (statistically not significant), while it strongly enhanced the internalization by Huh7 cells (Figure 6B,C) with a ~8.5-fold increase, compared to DL633-SA-Biot (Figure 6D). These data demonstrated that CPB peptide significantly reduced the difference in the cell uptake between the two cell types, although the internalization by HPM remained higher, since the HPM/Huh7 MFI ratio was ~3.2 (Figure 6C). Importantly, the uptake of CPBscr-DL633-SA bioconjugate was not significantly different to that of control DL633-SA-Biot complex in both cell types (Figure 6C,D), confirming that the specific amino acid sequence of CPB was crucial in the hepatoma cell targeting.

The uptake of native GBVA10-9-DL633-SA bioconjugate was much higher than those measured with DL633-SA-Biot control and CPB-DL633-SA complex in both cell types (Figure 6B,C). However, GBVA10-9 induced a much higher increase in complexes internalization in Huh7 (~384-fold), compared to that in HPM (~32-fold) (Figure 6D), which led to a 1.9 ratio value in MFI between the two cell types (Figure 6C). Interestingly, GBVA10-13 and GBVA10-9scr peptides also induced an increase in the uptake of bioconjugates in Huh7 and HPM, although significantly lower -MFI ration of 1.6) than that observed for GBVA10-9 (Figure 6C).

Together, these data demonstrated that both CPB and GBVA-10-9 peptides strongly enhanced the cell uptake of streptavidin in Huh7 hepatoma cells over that in human macrophages, and constitute good candidate ligands for targeting HCC. In addition, the CPB-mediated uptake of streptavidin relies on the specific amino acid sequence of the peptide, since its scrambled control counterpart has no effect on the streptavidin’s internalization in both cell types. In contrast, GBVA10-13 and GBVA10-9scr peptide complexes, although internalized in a lower extend that GBVA-10-9 conjugate, were much more internalized than the DL633-SA-Biot control, suggesting that the uptake of these three peptide bioconjugates by hepatoma cell does not strictly rely on the amino acid sequence of the peptides and the recognition of a cell surface receptor.

In order to determine the mechanism(s) involved in the uptake of peptide-streptavidin complexes by HPM, we studied the effects of phagocytosis and endocytosis inhibitors, latrunculin and genistein, respectively, on the internalization of control DL633-SA, CPB-, and GBVA-10-9-DL633-SA bioconjugates in the coculture model (Figure 7A).

The uptake of control- and CPB-DL633-SA complexes was significantly reduced by both latrunculin and genistein in HPM. In contrast, latrunculin had no effect on the uptake in Huh7 while genistein decreased the internalization of CPB-DL633-SA complex. Importantly, in cocultures exposed to latrunculin, the MFI in both HPM and Huh7 was nearly identical (MFI ratio at 1.1) (Figure 7B) demonstrating that pharmacological inhibition of phagocytosis in HPM improved the targeting of hepatoma cells by CPB peptide bioconjugates over the internalization by HPM.

As observed for CPB-DL633-SA complex, the treatment with latrunculin had no effect on the MFI in Huh7 while it unexpectedly increased the uptake of GBVA-10-9-DL633-SA bioconjugates in HPM (Figure 7). To further studied the internalization of GBVA-10-9-DL633-SA complex in HPM treated with latrunculin, we incubated GBVA-10-9-DL633-SA bioconjugate in pure culture of HPM (without Huh7 hepatoma cells), which confirmed the strong internalization of this complex in HPM in presence of latrunculin without changes in the distribution of the fluorescence staining (Appendix A). In contrast, the exposure to genistein decreased the MFI levels emitted by GBVA-10-9-DL633-SA complex in both cell types. 

These data strongly suggest that CPB-DL633-SA complex is internalized mainly by endocytosis in hepatoma cells and by a dual mechanism involving both phagocytosis and endocytosis in HPM. In addition, the very strong uptake of GBVA-10-9-DL633-SA bioconjugate occurs partially by endocytosis and probably by another mechanism, at least in HPM that could rely on physicochemical properties of this amphipathic helical peptide.

## 4. Discussion

Using peptides previously known to bind to human hepatoma cells in vitro and in xenograft HCC mouse models [10,49,50,51,52,53,55,57,63,64,65,66,67,68,69,70,71,72,88,89,90], we first determined which peptides exhibited the strongest tropism towards hepatoma cells by investigating the cell uptake of peptide bioconjugates and peptide-decorated NPs by different cell types, including hepatoma cell lines, human hepatocytes, and macrophages in primary cultures. The first striking result, regarding the internalization of peptide-fluorescent streptavidin DyLight^TM^ (488 DL488-SA) complexes, was that most selected peptides did not significantly enhance the binding and/or internalization of the fluorescent streptavidin by hepatoma cells. Peptides L5-2, CGK, SP94, and SRIF-14 showed weak positive effects on cell uptake, but not in all hepatoma cells. Conversely, CPB and GBVA10-9 peptides, derived from proteins expressed in pathogens infecting hepatocytes, strongly enhanced the uptake of DL488-SA in hepatoma cells and, to a much lesser extent, in human hepatocytes and NCI-H727 bronchial cancer cells. In contrast with the limited number of peptides affecting cell uptake of DL488-SA, the functionalization of Biot-PEG_62_-*b*-PMLABe_73_-based NPS [56,81,91], with all the selected peptides, especially CPB and GBVA10-9 peptides, strongly increased the NP’s uptake by hepatoma cells. These data further reinforce our strategy to develop route of synthesis to covalently bind peptides onto PEG_62_-*b*-PMLABe_73_ copolymers [92].

The striking difference observed in the effects of peptides on cell internalization between peptide-DL488-SA conjugates and peptide-functionalized NPs led us to question whether the specific amino acid sequence of the peptides was a crucial determinant in the cell uptake of peptide bioconjugates and peptide-decorated NPs. The concept of peptide-drug conjugates and peptide-functionalized NPs relies on the specific recognition by peptide ligands of membrane receptor(s) expressed by a subset of cells, in order to achieve an “active” cell targeting [3]. To investigate the role of the CPB and GBVA10-9 specific amino acid sequences in the cell uptake of streptavidin conjugates and peptide-decorated NPs, we used modified peptides CPBscr, GBVA10-13, and GBVA10-9scr, which were designed by redistribution or substitution in the native amino acid sequences. Experiments performed using native and modified peptide-DL488-SA demonstrated that cellular uptake of peptide-conjugates was significantly higher with CPB and GBVA10-9 than with their modified counterparts. These data indicated that the specific amino-acid sequence of CPB and GBVA10-9 peptides was a main determinant of the cell internalization for these bioconjugate models. In contrast, evaluation of cell uptake of peptide-functionalized NPs demonstrated that both GBVA10-9 and its scramble-control peptides strongly enhanced internalization by hepatoma cells, suggesting that peptides physicochemical features and/or formation of a protein corona could be the main drivers of the NPs’ cell internalization, rather than the recognition of a membrane receptor mediated by a specific peptide sequence, unlike the CPB peptide, for which the amino acid sequence is determinant for the cell uptake.

The opsonization is part of the innate immune system, enhancing the activity of the mononuclear phagocyte system (MPS) to clear pathogens from the body. This well-documented phenomenon, observed with all nano-objects, is a rapid process, leading to the formation of a protein corona at the surface of nanovectors, with a dynamic remodeling over time in blood [78,79,80,81]. Proteomics analyses performed on various NPs incubated with serum demonstrated that numerous plasma proteins (>100 different proteins and peptides) bind onto nano-vectors, including immunoglobulins, lipoproteins, complement, coagulation factors, and many others [78,79,80,81,93]. Authors have brought evidence that the protein corona formation and dynamics might depend on the physicochemical features of the nanovectors, including rigidity, hydrophilic, or hydrophobic surface, as well as charged or neutral zeta potential [93]. Proteins interact with NPs by different mechanisms, mainly electrostatic, hydrophobic, and hydrogen-bonding [93]. Lipoproteins, especially apolipoproteins E and A-I, are particularly enriched in NP’s protein corona, most likely through direct interaction with lipids, in case of liposomes and lipid NPs [94,95], as well as the hydrophobic core of polymeric NPs [78,79,80,81]. Interestingly, reports have recently shown that proteins within the corona, including the ApoE, induce structural rearrangements that may impact their cell internalization and overall biodistribution in vivo [93].

In order to reduce opsonization and scavenging by MPS, NPs have been optimized through the modulation of their size, shape [2], surface charge [96], and chemical structure [97,98,99], including the synthesis of amphiphilic copolymers with a poly(ethylene glycol) (PEG) block. The corresponding PEGylated NPs exhibit a hydrophobic core and hydrophilic corona of PEG segments that acts as a steric shield reducing the binding of opsonins and uptake by MPS and extending their systemic lifetime [78,79], but strongly reducing their interaction with most cell types and their subsequent endocytosis [91]. We previously demonstrated that the amphiphilic block copolymer Biot-PEG_62_-*b*-PMLABe_73_ also strongly reduced the formation of the protein corona and cell uptake by hepatoma cells, compared to the opsonization of NPs prepared from PMLABe_73_ homopolymer [81]. This observation was the rationale for the use of the Biot-PEG_62_-*b*-PMLABe_73_, in order to reduce the influence of the protein corona on the NP’s uptake.

The hypothesis that formation of a protein corona onto GBVA10-9 and CPB functionalized PEGylated NPs could be a crucial determinant of the cell uptake by hepatoma cells was investigated. Herein, we showed that addition of these two peptides modified the protein corona onto NPs by increasing the binding of several plasma proteins, including ApoA-I and ApoE, known to be main components of the protein corona of many polymeric nanomaterials [80]. ApoA-I being a major component of HDL, we evaluated the effect of purified HDL particles on the cell uptake of control biotin–streptavidin biotin–PEG_62_-*b*-PMLABe_73_ NPs, CPB-, and GBVA10-9 streptavidin–biotin-PEG_62_-*b*-PMLABe_73_ NPs. Addition of HDL to the culture medium of HepaRG hepatoma cells strongly reduced the cell uptake of peptide-functionalized NPs, but not that of control NPs. These data strongly suggested that the increase in cell internalization of GBVA10-9 and CPB functionalized NPs would involve the binding of HDL to these peptides. This conclusion was strengthened by pull-down assays using streptavidin–biotin-PEG_62_-*b*-PMLABe_73_ NPs and streptavidin-agarose beads harboring GBVA10-9 and CPB peptides, which further evidenced strong binding of ApoA-I and ApoE but also iC3b fragment, albumin, transferrin, and immunoglobulins to these two peptides. These data confirmed that GBVA10-9 and CPB peptides modulated the binding of plasma proteins to NPs and suggested that peculiar protein coronas characterized each peptide-functionalized NP to regulate cell uptake. Our data are in agreement with the previous reports showing that high affinity ApoA-I and ApoE to low-density lipoprotein (LDLR) and HDL receptors, which are overexpressed in liver cancer cells, could be used to target HCC [100,101,102].

In this study, we have also compared the in vitro uptake of CPB and GBVA10-9 peptides–streptavidin bioconjugates between hepatoma cells and macrophages, in order to determine whether these peptides endowed streptavidin bioconjugates with improved hepatoma cell targeting ability over that of macrophages, which are part of the mononuclear phagocyte system (MPS). The MPS in the normal liver involves resident Kupffer cells, while, in HCC, both Kupffer cells and monocyte-derived macrophages recruited to the tumor are very active [103], which impairs the targeting of tumoral hepatocytes by nanotechnology-based therapy. To address this issue, we developed an in vitro coculture system associating GFP positive Huh7 human hepatoma cells and primary macrophages (HPM) to mimic the cell competition for the internalization of our peptide-streptavidin bioconjugates. Using this model, we first evidenced that HPM were functional in this coculture system, since they exhibit a 22-fold higher phagocytic activity of control biotin-DL488-SA complex without peptide, compared to the internalization of the same control conjugate in Huh7 hepatoma cells. We next demonstrated that both CPB and GBVA10-9 peptides strongly improved the hepatoma cell targeting, possibly by different behaviors.

CPB peptide induced a ~10-fold increase in cell internalization in hepatoma cells, compared to a slight increase in HPM, while GBVA10-9 peptide strongly enhanced streptavidin cell uptake in both cell types. Peptide-DL488-SA conjugates were internalized both by phagocytosis and endocytosis, at least in part by a calveolin-dependent mechanism, since the genistein inhibitor of pinocytosis [74] reduced their cell uptake in HPM and hepatoma cells. However, differences in the levels of cell uptake and sensitivity to chemical inhibitors were observed. A possible explanation would be that the two peptides are internalized by different mechanisms in HPM and Huh7 cells. This hypothesis is reinforced by the following observations: (i) CPB-DL633-SA complex is internalized mostly by endocytosis in hepatoma cells and by both phagocytosis and endocytosis in HPM; (ii) genistein reduces GBVA10-9-DL488-SA’s uptake in both HPM and Huh7 cells, while the phagocytosis inhibitor latrunculin increased its internalization only in HPM. From these data and a previous report demonstrating that GBVA10-9 can inhibit HCV entry at a post-attachment step through its amphipathicity and helicity features [70], it can be hypothesized that the amphipathic helical GBVA10-9 peptide could interact with apolipoproteins and/or act as a cell penetrating peptide, both properties contributing to GBVA10-9-DL488-SA cell internalization, especially in HPM, independently from active phagocytosis and endocytosis. 

## 5. Conclusions

Our results showed that CPB and GBVA10-9, derived from the circumsporozoite protein of *Plasmodium berghei* and George Baker virus A, respectively, strongly increased the uptake of both streptavidin conjugates and biotin poly(ethylene glycol)-*block*-poly(benzyl malate) NPs in human hepatoma cells. In addition, our data provide evidence that peptide-functionalized nanomedicine may favor the uptake by hepatoma cells over that of macrophages and demonstrated that CPB- and GBVA10-9 constituted promising potent targeting agents for the diagnosis and/or treatment of HCC.

## Figures and Tables

**Figure 1 pharmaceutics-14-00804-f001:**
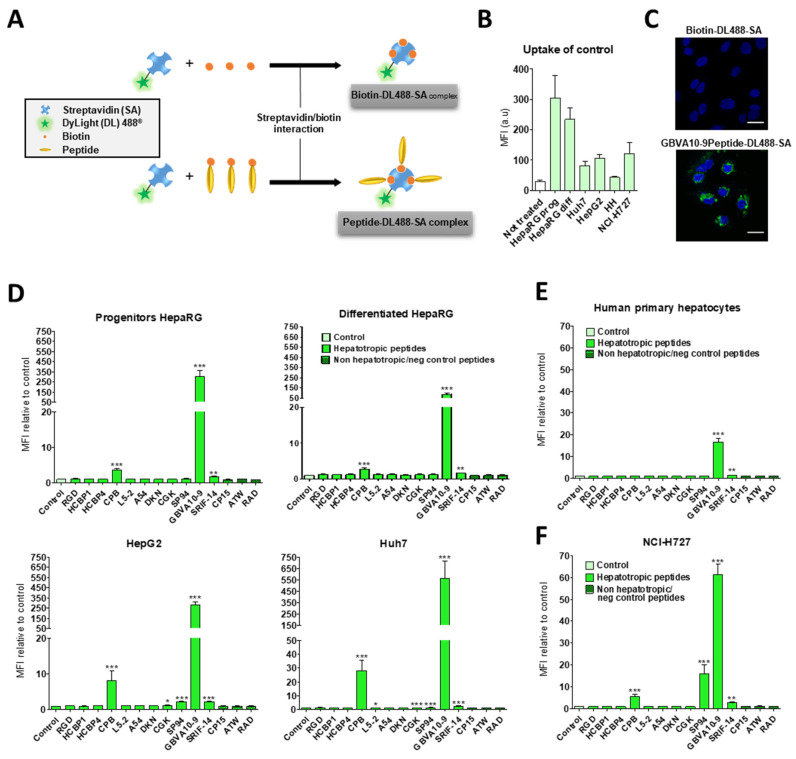
In vitro screening of peptide-fluorescent streptavidin conjugates. (**A**) Streptavidin DyLight 488^®^ (DL488-SA) was conjugated to biotin or biotinylated peptides to generate control- and peptide-streptavidin complexes, respectively. The cell uptake of DL488-SA was quantified after 24 h incubation by flow cytometry with FITC channel, to measure the relative mean of cell fluorescence (MFI). (**B**) Comparative cell uptake of the control biotin-DL488-SA conjugate by hepatic cancer cell lines (HepaRG, HepG2, and Huh7), human primary hepatocytes (HH), and bronchial NCI-H727 cells. The laser’s voltage was adjusted to bring all untreated cells at MFI of 30 arbitrary units (a.u). Data are presented as the mean of fluorescence intensity (MFI) ± SD. (**C**) Fluorescence microscopy images of differentiated HepaRG after 24 h incubation, with control or biotinylated-GBVA10-9 peptide conjugated to DL488-SA. Bars correspond to 20 µm. In vitro uptake of peptide- DL488-SA conjugates in (**D**) hepatic cancer cell lines, (**E**) human primary hepatocytes, and (**F**) bronchial carcinoid cell line. In order to evaluate the effect of peptides on the cellular uptake, data are presented as the mean of fluorescence intensity (MFI) of cells incubated with peptide-streptavidin complexes, relative to MFI in cells incubated control (biotin–streptavidin) conjugate ± SD (MFI ratio). Significant changes in MFI ratio at * *p* ≤ 0.05, ** *p* ≤ 0.01, *** *p* ≤ 0.001, control vs. peptides. Statistical analyses were performed using Kruskal–Wallis non-parametric test, followed by Dunn’s post-hoc test.

**Figure 2 pharmaceutics-14-00804-f002:**
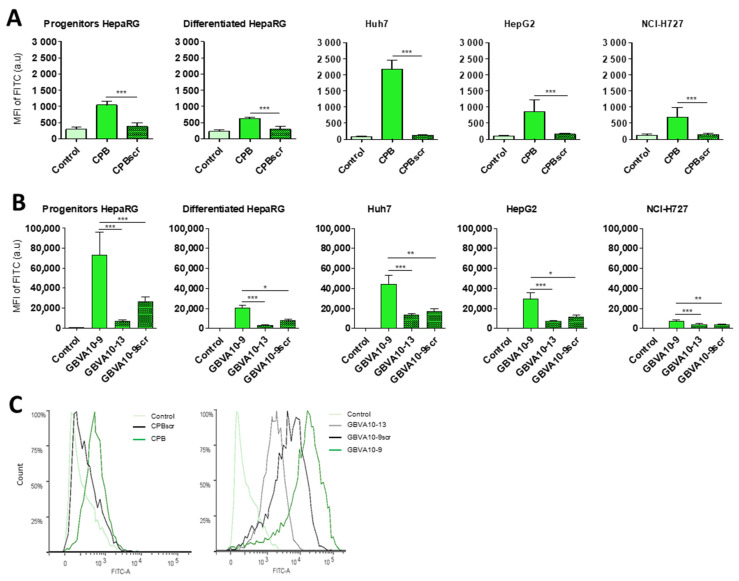
Influence of the amino acid sequence of CPB and GBVA10-9 on the uptake of peptide-streptavidin complexes. CPB, GBVA10-9, and their mutated/scrambled controls were conjugated to streptavidin DyLight 488^®^ (DL488-SA). Cell uptake was evaluated by flow cytometry after 24 h incubation with hepatic cell lines (HepaRG, Huh7, and HepG2) or bronchial NCI-H727 cells. MFI were first adjusted in untreated cells (control) and arbitrary set at ≈30 units (a.u) for all cell types. (**A**) Comparison between CPB and its scrambled peptide. Statistical analyses were performed using Mann–Whitney non-parametric test: *** *p* ≤ 0.001, CPB vs. CPBscr. (**B**) Comparison between GBVA10-9 and its control peptides (GBVA10-13 and GBVA10-9scr). Statistical analyses were performed using Kruskal–Wallis non-parametric test, followed by Dunn’s post-hoc test: ** p* ≤ 0.05, ** *p* ≤ 0.01, *** *p* ≤ 0.001, GBVA10-9 vs. control peptides. (**C**) Histograms of flow cytometry illustrating the GBVA10-9-, GBVA10-13, and GBVA10-9scr-DL488-SA uptake by differentiated HepaRG cells.

**Figure 3 pharmaceutics-14-00804-f003:**
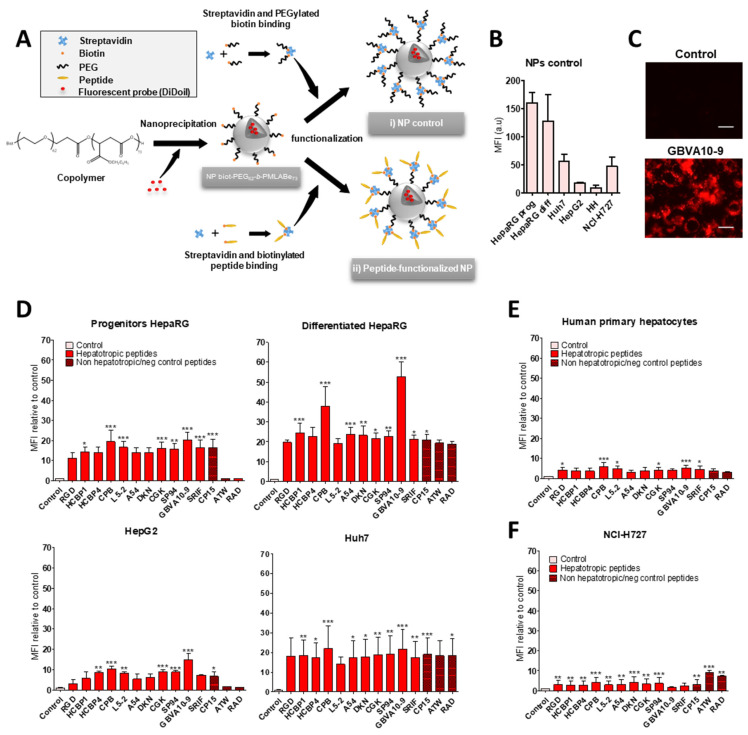
Uptake of peptide-functionalized NPs. (**A**) The fluorescent probe DiD Oil was encapsulated into NPs prepared from Biot-PEG_62_-*b*-PMLABe_73_ copolymer to allow their detection. NPs were: (i) control biotin–streptavidin (Biot-Strept) Biot-PEG_62_-*b*-PMLABe_73_ NPs; (ii) peptide-functionalized Biot-PEG_62_-*b*-PMLABe_73_ NPs using biotinylated-peptide streptavidin bridging. Comparative cell uptake was measured after 24 h incubation by flow cytometry on APC channel. (**B**) Comparative cell uptake of control Biot-Strept-Biot-PEG_62_-*b*-PMLABe_73_ NPs by hepatic cancer cell lines (HepaRG, HepG2, and Huh7), human primary hepatocytes (HH), and bronchial NCI-H727 cells. The mean of fluorescence intensity (MFI) in untreated cells was arbitrary set at ≈30 units (a.u) for all cell types. (**C**) Fluorescence microscopy images of differentiated HepaRG after 24 h incubation, with control or GBVA10-9 peptide conjugated to NPs. Bars correspond to 100 µm. Effects of the different peptides on the NP’s uptake in (**D**) hepatoma cells, (**E**) human primary hepatocytes, and (**F**) bronchial carcinoid cells after 24 h incubations. Data are presented as the ratio of the mean of fluorescence intensity (MFI) of cells incubated with peptide-functionalized NPs relative to MFI in cells incubated with control Biot-Strept-Biot-PEG_62_-*b*-PMLABe_73_ NPs ± SD (MFI ratio). Statistical analyses were performed using Kruskal–Wallis non-parametric test, followed by Dunn’s post-hoc test: * *p* ≤ 0.05, ** *p* ≤ 0.01, *** *p* ≤ 0.001, control vs. peptides NPs.

**Figure 4 pharmaceutics-14-00804-f004:**
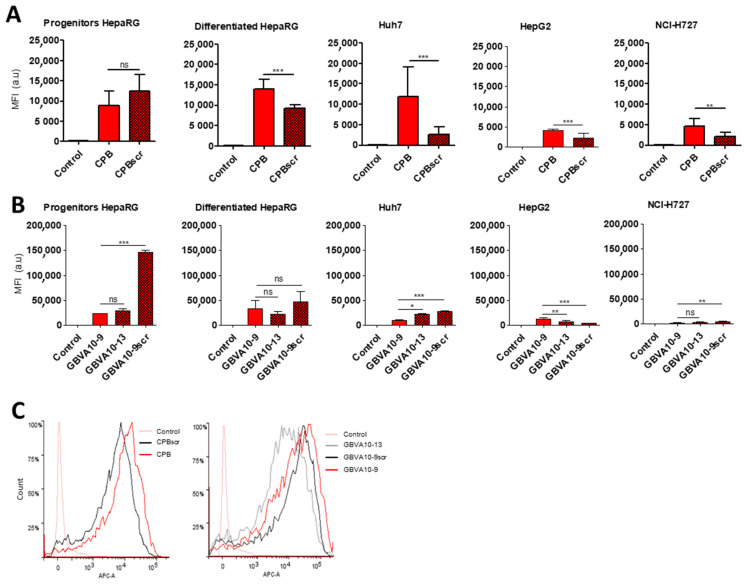
Influence of the amino acid peptide sequence of CPB and GBVA10-9 in the uptake of functionalized NPS. CPB, GBVA10-9 and their modified (mutated/scrambled) control peptides were grafted on NPs via streptavidin/biotin bridge. Cell uptake of NPs was evaluated by flow cytometry (APC channel) after 24 h incubation in hepatic cancer cell lines (HepaRG, Huh7, and HepG2) or bronchial NCI-H727 cells. Data are presented as mean of fluorescence intensity (MFI) ± SD. MFI were first adjusted in untreated cells (control) arbitrary set at 30 units (a.u) for all cell types. (**A**) Comparison between CPB and its scrambled peptide. Statistical analyses were performed using Mann–Whitney non-parametric test: ^ns^ not significant, ** *p* ≤ 0.01, *** *p* ≤ 0.001, CPB vs. CPBscr. (**B**) Comparison between GBVA10-9 and its control peptides (GBVA10-13 and GBVA10-9scr) Statistical analyses were performed using Kruskal–Wallis non-parametric test, followed by Dunn’s post-hoc test: ^ns^ not significant, * *p* ≤ 0.05, ** *p* ≤ 0.01, *** *p* ≤ 0.001, GBVA10-9 vs. negative control peptides. (**C**) NPs uptake by differentiated HepaRG cells visualized by flow cytometry. Histograms of flow cytometry illustrating the uptake of GBVA10-9-, GBVA10-13, and GBVA10-9scr-functionalized NPs in differentiated HepaRG cells.

**Figure 5 pharmaceutics-14-00804-f005:**
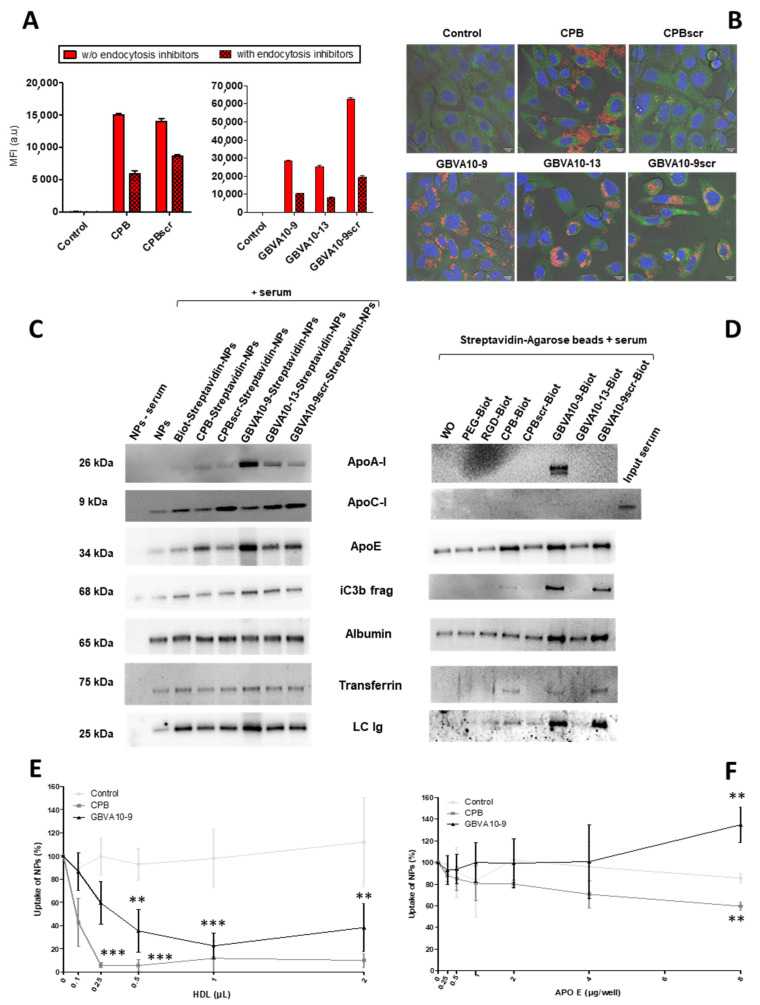
Mechanism of NP’s uptake in HepaRG cells (**A**) Internalization (MFI) of NPs functionalized with CPB, GBVA10-9 and their modified (mutated/scrambled) peptides by HepaRG cells in absence or presence of the endocytosis inhibitors genistein and chlorpromazine, measured by flow cytometry. ((**B**) and Appendix A) Confocal microscopy of progenitor HepaRG cells incubated with CPB, GBVA10-9, and their modified (mutated/scrambled) peptides for the detection of the endosomal Rab7 GTPase (green), fluorescent nanocarriers containing DiD Oil (red) and the nuclear DNA stained with Hoechst (blue). (**C**) Immunoblotting of human plasma proteins [apolipoproteins (Apo) AI, CI, E, IC3b complement fragment (IC3b), albumin, transferrin, and light chain of immunoglobulins (LC Ig)] bound onto bare NPs and peptide-functionalized NPs. (**D**) pull-down assay and immunoblotting of human plasma proteins using bare and peptide-functionalized agarose beads. Internalization of fluorescent control NPs, CPB, and GBVA10-9 peptide-functionalized NPs by HepaRG cells, in the absence or presence of increasing amounts of Human HDL (**E**) and recombinant apolipoprotein E (**F**). Fluorescence intensity (MFI) was arbitrarily set as 100% in cells incubated with control and peptide-functionalized NPs in absence of HDL or ApoE. Variations in MFI of cells incubated with NPs in presence of HDL and ApoE were expressed in percentage of the corresponding control condition. Statistical analyses were performed using Kruskal–Wallis non-parametric test, followed by Dunn’s post-hoc test: ** *p* ≤ 0.01, *** *p* ≤ 0.001, GBVA10-9-, or CPB-functionalized NPs vs. control Biot-PEG_62_-*b*-PMLABe_73_ NPs.

**Figure 6 pharmaceutics-14-00804-f006:**
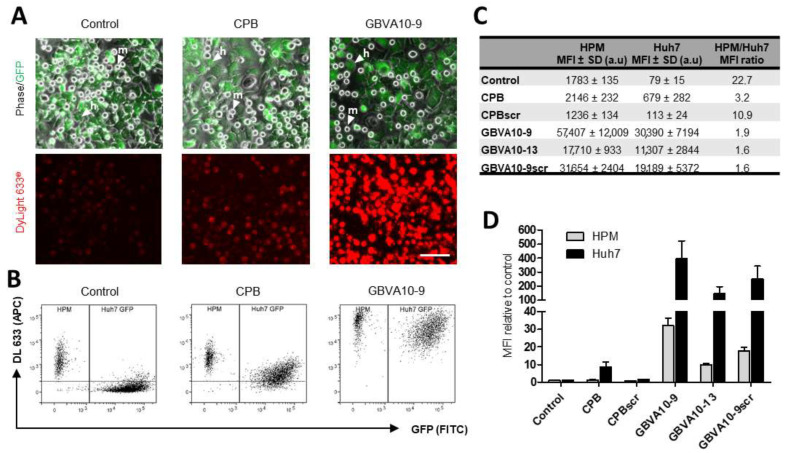
Coculture between human primary macrophages (HPM) and Huh7 cells. HPM- and GFP-expressing Huh7 cells were cultured together in same well during 24 h. Then streptavidin DyLight 633^®^-peptide conjugates (DL633-SA-peptide) were added in culture medium for 24 h incubation. Cells were wash twice before analysis. Cellular uptake of control DL633-SA-Biot and DL633-SA-peptide was evaluated by fluorescence microscopy and flow cytometry. (**A**) Upper panel: HPM (m) are visualized by phase contrast and GFP expressing Huh7 cells (h) are detected by fluorescence microscopy at 470 nm. Lower panel: Intracellular DL633-SA-Biot and DL633-SA-peptide were observed by fluorescence microscopy at 590 nm. Images shown are representative of multiple independent experiments. White bar corresponds to 10 µm. (**B**) Representative flow cytometry dot plots. The rectangular gates separate HPM on the left and GFP-expressing Huh7 on the right, according to fluorescence detection using FITC channel. We analyzed DL633-SA-Biot and DL633-SA-peptide uptake in each cell populations by measuring fluorescence intensity on APC channel. (**C**) Data are presented as mean of fluorescence intensity on APC channel ± SD. In the last column, we calculated the ratio between the fluorescence emitted by HPM and Huh7. (**D**) Peptides effect on the uptake of streptavidin Dylight by HPM and Huh7 in coculture. Data are presented as the mean of fluorescence intensity (MFI) relative to control ± SD.

**Figure 7 pharmaceutics-14-00804-f007:**
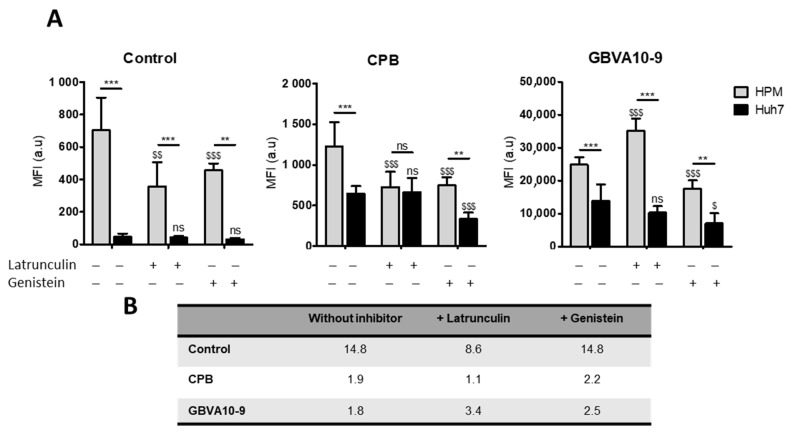
Influence of phagocytosis or endocytosis inhibition on DL633-SA-peptide bioconjugates uptake by cocultured HPM/Huh7. Cocultures of HPM and GFP-transfected Huh7 were treated with control or DL633-SA-peptide bioconjugates in absence or presence of phagocytosis inhibitor (latrunculin 500 nM) or endocytosis inhibitor (genistein 100 µM). After 4 h of incubation at 37 °C, cells were washed twice and the cell uptake of bioconjugates was evaluated by flow cytometry. (**A**) Data are presented as mean of fluorescence intensity (MFI) on APC channel ± SD. Statistical analyses were performed using Mann–Whitney non-parametric test. ^ns^ not significant, ** *p* ≤ 0.01, *** *p* ≤ 0.001, HPM vs. Huh7. ^ns^ not significant, ^$^
*p* ≤ 0.05, ^$$^
*p* ≤ 0.01, ^$$$^
*p* ≤ 0.001, comparison with the corresponding condition without inhibitor treatment. (**B**) In the table, we calculated the ratio between the fluorescence emitted by HPM and the fluorescence emitted by Huh7.

## Data Availability

Data supporting reported results are publicly available at Mendeley data (https://data.mendeley.com/) and can be accessed (since 1 April 2022) using the following link: Loyer, Pascal (2022), “Vène et al. CPB- and GBVA-derived peptides trigger efficient cell internalization of bioconjugates and functionalized PMLABe-based nanoparticles in human hepatoma cells”, Mendeley Data, V1, doi: 10.17632/tcx26rtxf8.1.

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
