# Peer review of "Circumsporozoite Protein of Plasmodium berghei- and George Baker Virus A-Derived Peptides Trigger Efficient Cell Internalization of Bioconjugates and Functionalized Poly(ethylene glycol)-b-poly(benzyl malate)-Based Nanoparticles in Human Hepatoma Cells"

_pharmaceutics, 2022, doi:10.3390/pharmaceutics14040804_

Round 1

Reviewer 1 Report

I have a few points (see below) that need to be addressed before publication.

The figures are very difficult to read and should be of much higher quality.

How was the mean fluorescence intensity (MFI) calculated? A description should be provided as it is main part of results.

The authors should explain how proteins such as lipoproteins get on the surface of particles, affecting their cellular internalization. Is this interaction specifically due to binding or charge interactions?

Clear explanation of figure 7 should be provided. For example for CPB it seems endocytosis is main mechanism for Huh7, while for HPM it is both phagocytosis and endocytosis. For GBVA10-9 it seems both phagocytosis and endocytosis for Huh7. It is unclear why MFI increases with latrunculin? Possible reason(s) should be provided. Maybe fluorescence imaging experiments can provide insights, similar to those presented.

Since the authors mention particles for theranostics they should provide a few examples in the manuscript where NPs can be used for theranostics- both imaging and therapy (see examples below).

[1] Fernandes DA, Kolios MC. Near-infrared absorbing nanoemulsions as nonlinear ultrasound contrast agents for cancer theranostics. Journal of Molecular Liquids. 2019;287:110848.

[2] Fernandes DA, Fernandes DD, Malik A, Gomes G-NW, Appak-Baskoy S, Berndl E, et al. Multifunctional nanoparticles as theranostic agents for therapy and imaging of breast cancer. Journal of Photochemistry and Photobiology B: Biology. 2021;218:112110.

Reviewer 2 Report

Authors investigated and described the ability of several peptides, selected from the literature, to target human hepatoma cells. They performed a deep investigation based on cell uptake assays by using biotinylated synthetic peptides bound to fluorescent streptavidin or conjugated onto nanoparticles (NPs) exploiting the streptavidin bridging.

The manuscript is clear, well-written and it is a topic of interest to the researchers in the related areas. It is suitable for a publication in Pharmaceutics; however, figures have a very low resolution and they are unreadable, moreover axis of graphs and legends are too small.

Fluorescent and confocal images are not clear.

Figures quality must be improved before publications.

Minor comments are as follows:

  • chlorpromazine is not mentioned in Methods section.
  • the nanoparticles concentration used for the different tests should be indicated.

Reviewer 3 Report

In this manuscript, biotinylated synthetic peptides bound to fluorescent streptavidin or engrafted onto Biot-PEG-b-PMLABe -nanoparticles were analyzed to tropism towards human hepatoma cells. CPB and GBVA10-9 peptide-streptavidin conjugates and nanoparticles functionalized showed favored apolipoprotein binding to hepatoma cells over that of human macrophages.

In my point-of-view, the manuscript was well designed including appropriate methods, relevant results and discussion, and a conclusion reporting the main findings.

Minor comments are about figure resolutions that show a very low quality making a proper analysis difficult.

Round 2

Reviewer 1 Report

The manuscript is now acceptable for publication.